# Adapting Prediction Sets to Distribution Shifts Without Labels

## Abstract

Recently there has been a surge of interest to deploy confidence set predictions rather than point predictions. Unfortunately, the effectiveness of such prediction sets is frequently impaired by distribution shifts in practice, and the challenge is often compounded by the lack of ground truth labels at test time. Focusing on a standard set-valued prediction framework called conformal prediction (CP), this paper studies how to improve its practical performance using only unlabeled data from the shifted test domain. This is achieved by two new methods called `ECP` and `EACP`, whose main idea is to adjust the score function in CP according to its base model's own uncertainty evaluation. Through extensive experiments on a number of large-scale datasets and neural network architectures, we show that our methods provide consistent improvement over existing baselines and nearly match the performance of fully supervised methods.

## 1 Introduction

Advances in deep learning are fundamentally changing the autonomous decision making pipeline. While most works have focused on accurate point predictions, quantifying the uncertainty of the model is arguably as important. Taking autonomous driving for example: if a detection model predicts the existence of an obstacle, it would be reasonable to take different maneuvering strategies depending on the confidence of the prediction. But is that reliable? In a possible failure mode, the model could report $60\%$ (resp. $99\%$) confidence, but the *probability* of an obstacle actually showing up is $99\%$ (resp. $60\%$). Such discrepancy between the model's own uncertainty evaluation and the ground truth probability (or post-hoc frequency) is actually very common (Guo et al., 2017; Liang et al., 2023), and can significantly compromise the safety in downstream decision making.

Set-valued prediction provides an effective way to address this problem (Chzhen et al., 2021), with *conformal prediction* (CP; Vovk et al., 2005) being its most well-known special case. Given a fixed black-box machine learning model (called the *base model*) and a covariate $x_{\text{test}}$, the goal of CP is to generate a small prediction set $\mathcal{C}_{\text{test}}$ that contains (or, *covers*) the unknown ground truth label $y_{\text{test}}$ with a pre-specified probability. Crucially, CP relies on the assumption that the distribution of the data stream is *exchangeable* (a weaker variant of i.i.d.), which allows the fairly straightforward inference of $y_{\text{test}}$ from $x_{\text{test}}$ and the base model's performance on a pre-collected *calibration dataset*. Note that the ground truth label $y_{\text{test}}$ does not need to be revealed after the set prediction is made: exchangeability together with a large enough labeled calibration dataset is sufficient to ensure the desirable coverage probability. This is particularly important for autonomous decision making, where real-time data annotation is expensive or even infeasible.

However, real-world data streams are usually corrupted by all sorts of *distribution shifts*, violating the exchangeability assumption. Even when the data stream itself is exchangeable, we often want to continually update the base model rather than keeping it fixed, and this can be effectively understood as a distribution shift in this context. In such cases, simply applying exchangeability-based CP methods could lead to highly inaccurate prediction sets (Tibshirani et al., 2019; Bhatnagar et al., 2023a; Kasa and Taylor, 2023). Therefore, making CP compatible with distribution shifts has become a focal point of recent works.

A number of solutions have been proposed, but the key challenge still remains. For example, Gibbs and Candes (2021) formulated the connection between CP and *Online Convex Optimization* (OCO; Zinkevich, 2003), and the latter is able to handle arbitrarily distribution-shifted environments. The

weakness is that ground truth labels are now required at test time (which we call *full supervision*), as opposed to the standard CP procedure. In the other direction, there are CP methods that combat distribution shifts without test time labels (Tibshirani et al., 2019; Barber et al., 2023; Cauchois et al., 2024), but they typically assume the distribution shifts are "easy", such that even without labels, we can still rigorously infer the test distribution to a certain extent using the labeled calibration dataset. Overall, it appears that handling both difficulties – distribution shifts and the lack of test time labels – is a formidable but important challenge remaining in the literature.

**Contributions** Focusing on classification, this paper develops practical unsupervised methods to improve the accuracy degradation of CP prediction sets under distribution shifts. The overarching idea is to exploit the uncertainty evaluation of the base model itself. Although such a quantity is not always calibrated in a strict sense, it has been consistently observed to strongly correlate with the magnitude of distribution shifts (Hendrycks and Gimpel, 2017; Wang et al., 2021; Kang et al., 2024), thus providing a valuable way to probe the test distribution without label access. Under this high level idea, we make the following contributions.

- First, we propose a new CP-inspired method named ECP (Entropy scaled Conformal Prediction). The key idea is to scale up the *score function* in standard CP by an "entropy quantile" of the base model, calculated on the unlabeled test dataset. Such an entropy quantile measures the base model's own uncertainty on the test distribution, and is enforced to be greater than 1.[1]

  More precisely, given each covariate $x_{\text{test}}$ at test time, the score function in standard CP is determined by the fixed base model, and assigns each candidate label a "propensity score". Then, the CP prediction set $\mathcal{C}_{\text{test}}$ simply includes all the candidate labels whose score is above a certain threshold.[2] By scaling up the score function while keeping the threshold fixed, ECP makes the prediction sets larger, which naturally corresponds to the intuition that the uncertainty of prediction should be inflated under distribution shifts. Moreover, the amount of such inflation is strongly correlated with the magnitude of the distribution shift, through the use of the entropy quantile.

- Second, we refine ECP using techniques from unsupervised *Test Time Adaptation* (TTA) (Niu et al., 2022), and the resulting method is named EACP (Entropy base-adapted Conformal Prediction). The key idea is that while ECP keeps the base model fixed at test time, we can concurrently update it using *entropy minimization* (Grandvalet and Bengio, 2004; Wang et al., 2021) – a widely adopted idea in unsupervised TTA, alongside the aforementioned entropy scaling. This "adaptively" reduces the scaling effect that ECP applies to the score function, thus shrinking the prediction sets of ECP smaller.

- Finally, we evaluate the proposed methods on a wide range of large-scale datasets under distribution shifts, as well as different neural network architectures. We find that exchangeability-based CP (with and without TTA on the base model) consistently leads to lower-than-specified coverage frequency. However, despite the absence of practical statistical guarantees in this setting, our methods can effectively mitigate this under-coverage issue while keeping the sizes of the prediction sets moderate. Furthermore, our methods also significantly improve the prediction sets generated by the base model itself (without CP). It shows that by bridging the CP procedure (which is statistically sound) and the base model's own uncertainty evaluation (which is often informative), our methods enjoy the practical benefit from both worlds.

## 2 RELATED WORKS

**CP under distribution shifts** Considerable efforts have been devoted to developing CP methods robust to distribution shifts, which can be approximately categorized into two directions. The first direction does not require test time labels (Tibshirani et al., 2019; Cauchois et al., 2024), but the distribution shift is assumed to be simple in some sense. The second direction is connecting CP to adversarial online learning (Gibbs and Candes, 2021), but the true labels are required at test time. Due to space constraints, a thorough discussion is deferred to Appendix A, as well as a number of applications that motivate this work.

**Unsupervised *Test Time Adaptation* (TTA)** Our techniques are inspired by core ideas in (unsupervised) TTA, whose goal is to update a trained machine learning model at test time, using unlabeled

---

[1]This is to ensure that the prediction sets do not become smaller on in-distribution data.

[2]The score functions are assumed to be *positively oriented* (Sadinle et al., 2019): labels with larger score are more likely to be included in the prediction set.

data from shifted distributions. To achieve this, one could update the batch-norm statistics on the test data (Nado et al., 2020; Schneider et al., 2020; Khurana et al., 2021), or minimize the test-time *prediction entropy* – a natural measure of the model's uncertainty (Wang et al., 2021; Zhang et al., 2022; Niu et al., 2022; Song et al., 2023; Press et al., 2024). Notably, these methods can be applied to any probabilistic and differentiable model (such as modern neural networks), which is naturally congruent with the key strength of CP. However, to date this line of works has not been connected to the conformal prediction literature.

## 3 PRELIMINARIES OF CP

We begin by introducing the standard background of CP without distribution shifts. For clarity, we assume i.i.d. data in our exposition, rather than the slightly weaker notion of exchangeability. Also see (Roth, 2022; Angelopoulos and Bates, 2023; Tibshirani, 2023).

Let $\mathcal{D}$ be an unknown distribution on the space $\mathcal{X} \times \mathcal{Y}$ of covariate-label pairs, and let $\alpha \in (0, 1)$ be the *error rate* we aim for. Given a calibration dataset $D$ consisting of $n$ i.i.d. samples $\{x_i^*, y_i^*\}_{i \in [n]} \sim \mathcal{D}^n$, the goal of CP is to generate a set-valued function $\mathcal{C} : \mathcal{X} \rightarrow 2^{\mathcal{Y}}$, such that

$$\mathbb{P}_{(x_{\text{test}}, y_{\text{test}}) \sim \mathcal{D}, D \sim \mathcal{D}^n} \left[ y_{\text{test}} \in \mathcal{C}(x_{\text{test}}) \right] \geq 1 - \alpha. \tag{1}$$

That is, for a fresh test sample $(x_{\text{test}}, y_{\text{test}}) \sim \mathcal{D}$, our prediction set $\mathcal{C}(x_{\text{test}})$ covers the ground truth label $y_{\text{test}}$ with guaranteed high probability. Notice that Eq.(1) alone is a trivial objective, since it suffices to predict the entire label space $\mathcal{C}(x) = \mathcal{Y}$ for all $x$. Therefore, CP is essentially a bi-objective problem: as long as Eq.(1) is satisfied, we want the prediction set $\mathcal{C}(x)$ to be small.

The main difficulty of this set-valued prediction problem is that the range of output $2^{\mathcal{Y}}$ is too large. In this regard, the key idea of CP is reducing the problem to 1D prediction via a trained machine learning model (called the *base model*), such as a neural network. Specifically, we assume access to a (positively oriented; i.e., larger is better) *score function* $s : \mathcal{X} \times \mathcal{Y} \rightarrow \mathbb{R}_+$ given by the base model, such that for each test covariate $x_{\text{test}} \in \mathcal{X}$ and *candidate label* $y \in \mathcal{Y}$, $s(x_{\text{test}}, y)$ measures how likely the model believes that $y$ is the true label $y_{\text{test}}$. Then, all there is left for CP is to pick a threshold $\tau_D \in \mathbb{R}$ that depends on the dataset $D$, and predict the label set (if the score function is negatively oriented, then $\geq$ is replaced by $\leq$)

$$\mathcal{C}(x_{\text{test}}) := \{y \in \mathcal{Y} : s(x_{\text{test}}, y) \geq \tau_D\}. \tag{2}$$

Under the i.i.d. assumption, the coverage objective Eq.(1) is satisfied by picking $\tau_D$ as the $\alpha(1 - n^{-1})$-quantile of the *empirical scores* $\{s(x_i^*, y_i^*)\}_{i \in [n]}$. Since the training data of the base model is split from the calibration dataset used to determine $\tau_D$, this approach is commonly known as *split conformal prediction*, which we refer to as `SplitCP`. Notably, $\tau_D$ is determined by the calibration dataset $D$; once the latter is fixed, there is no need to access the ground truth labels at test time.

**Examples in classification** This paper focuses on classification. In this case, a simple and popular choice of the score function is $s(x, y) = \pi_\theta(x)_y$ (Sadinle et al., 2019), where $\pi_\theta$ is a trained neural network parameterized by $\theta$, and $\pi_\theta(x)_y \in [0, 1]$ is the softmax score corresponding to one of the $k$-classes $y \in [k]$. Such a score function is positively oriented, which we adopt in this work. We note that another well-known choice due to Romano et al. (2020) is negatively oriented, and our methods can be applied there as well.

**Distribution shift** For the rest of this paper, we study the following deviation of the above standard CP problem. At test time, instead of working with a single test sample $(x_{\text{test}}, y_{\text{test}})$ drawn from $\mathcal{D}$, we consider a size-$N$ collection of samples[3] $\{x_i, y_i\}_{i \in [N]}$ drawn from some new unknown distribution $\mathcal{D}_{\text{test}}$. We only observe the covariates, defined as the *test dataset* $D_{\text{test}} = \{x_i\}_{i \in [N]}$. Importantly, the ground truth labels on $D_{\text{test}}$ are not revealed even after predictions are made. The goal, from a practical perspective, is to output a small prediction set $\mathcal{C}(x_i)$ at each test covariate $x_i$, satisfying the specified *empirical coverage rate*,

$$\frac{1}{N} \sum_{i=1}^{N} \mathbf{1}[y_i \in \mathcal{C}(x_i)] \geq 1 - \alpha.$$

---

[3]The clearest notation is to index the test samples by $(x_{\text{test},i}, y_{\text{test},i})$. Here we omit the subscript "test" for conciseness.

Notice that the function $\mathcal{C}$ can now depend on both the labeled calibration dataset $D$ and the unlabeled test dataset $D_{\text{test}}$.

In general, it is impossible to prove meaningful bounds without assuming some form of similarity between $\mathcal{D}$ and $\mathcal{D}_{\text{test}}$, but we will show that with more help from the base model, the CP procedure can be modified to work well in practice.

# 4 OUR METHODS

In this section, we first propose a method called ECP (Entropy scaled Conformal Prediction), which improves the coverage rate of CP by enlarging its prediction sets using the uncertainty evaluation of the base model itself. Crucially, this notion of uncertainty can be directly minimized and refined through unsupervised TTA, leading to an improved method called EACP (Entropy base-Adapted Conformal Prediction). The latter is able to both

- recover the desired error rate on many challenging distribution-shifted datasets; and
- significantly reduce inflated set sizes under increased uncertainty.

## 4.1 SCALING CONFORMAL SCORES BY UNCERTAINTY

Let us start with a high-level motivation. Within the SplitCP framework, an important design objective is *local adaptivity*: the size of the prediction set $\mathcal{C}(x)$ needs to vary appropriately with the covariate $x$. To this end, standard practice is to adjust the score function $s(x, y)$ based on some notion of uncertainty (or difficulty) that the base model decides at each $x$ (Papadopoulos et al., 2008; Johansson et al., 2015; Lei et al., 2018; Izbicki et al., 2019; Romano et al., 2019; Seedat et al., 2023; Rossellini et al., 2024). This has the effect of inflating the prediction set on the base model's uncertain regions, and has been shown to improve the more informative *conditional coverage rate* of CP (Angelopoulos and Bates, 2023; Tibshirani, 2023).

**Key idea** Inspired by these results, our key idea is to apply an analogous uncertainty scaling on the score function, in order to improve the performance of CP under distribution shifts. However, instead of using the uncertainty of the base model at each covariate $x$, we draw a crucial connection to unsupervised TTA, and evaluate the base model's uncertainty on the whole distribution-shifted *test dataset* $D_{\text{test}}$ – this effectively aggregates its "localized" uncertainty at the test covariates $\{x_i\}_{i \in [N]}$. In other words, instead of aiming for "local adaptivity" as in prior works, we use uncertainty scaling to achieve the adaptivity w.r.t. the unknown distribution shift.

**Prediction entropy** More concretely, which uncertainty measure should we use on the base model? As discussed above, the ideal dataset-specific uncertainty measure would follow from a "localized" uncertainty measure at each covariate $x$, and in the context of classification, a particularly useful one is the *entropy* of the base model's probabilistic prediction,

$$h(x) = -\sum_{y \in [k]} \pi_\theta(x)_y \log \pi_\theta(x)_y.$$

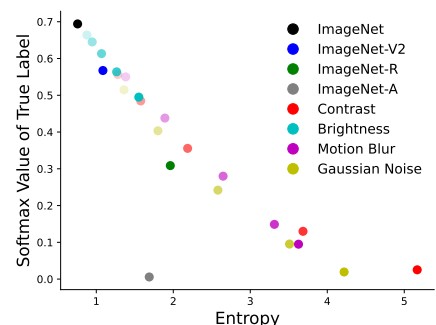

Figure 1: Entropy vs. the softmax score of the true label, averaged on each dataset. Different colors represent different datasets, and darker shades represent greater severity levels of ImageNet-C corruptions. See Section 5 for more details on the datasets.

Previous works have established the relation between such an entropy notion and the magnitude of the distribution shift, showing that larger shifts are strongly correlated with higher entropy (thus higher uncertainty in the base model), e.g., (Wang et al., 2021; Kang et al., 2024). We provide a consistent but unique observation in Figure 1, which plots the relation between the entropy (averaged over all $x$ values in the dataset) and the *softmax score of the true label* (also averaged over $x$), evaluated on a ResNet-50 model[4] and across a range of datasets. For the true label to be included in the CP

---

[4]We fix the base model to ResNet-50 in most of our experiments, unless otherwise specified.

prediction set, which is eventually what we aim for, its softmax score should be greater than the CP threshold $\tau_D$. Figure 1 shows that an increase in entropy is associated with a decrease in the softmax score of the true label, which crucially means that we need to scale up the score function in order to still cover the true label.

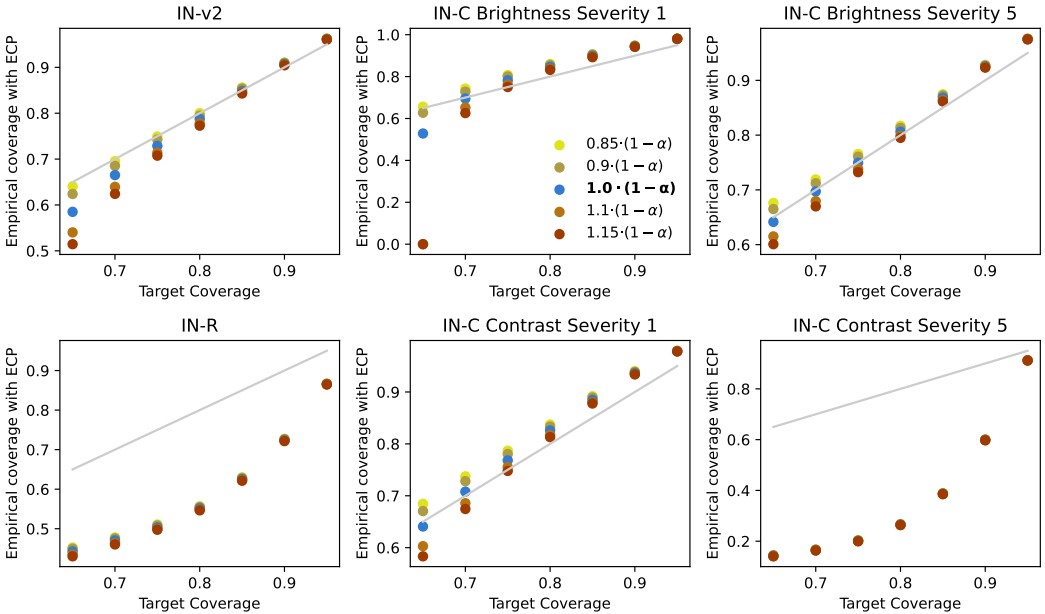

Figure 2: The targeted coverage rate $1 - \alpha$ vs. the empirical coverage rate, induced by ECP with different $\beta$ values (represented by different colors). It shows that simply setting $\beta = 1 - \alpha$ in ECP (i.e., the blue dots) consistently works well for all but the most severe distribution shifts (e.g., ImageNet-R and ImageNet-C Contrast Severity 5). Such an observation also holds across various $\alpha$ values, suggesting the effectiveness of this hyperparameter choice.

Now consider going from the "localized" uncertainty measure $h(x)$ to an uncertainty measure on the test dataset $D_{\text{test}}$, denoted as $u_{\text{test}}$. One could use the average $N^{-1} \sum_{i=1}^{N} h(x_i)$, but to increase the robustness, we define $u_{\text{test}}$ as the $\beta$-quantile of $\{h(x_i)\}_{i \in [N]}$, where $\beta$ is a hyperparameter. Quite surprisingly, we find that simply setting $\beta$ to the desired coverage rate $1 - \alpha$ is a fairly reliable choice in practice (see Figure 2), which gives a robust (over)-estimate of typical $h(x)$ values on the test dataset. We perform all the experiments with this direct relationship to avoid excessive hyperparameter tuning, but it can be further refined if desired.

**Method: ECP**   Now we are ready to use $u_{\text{test}}$ above to scale the score functions on the test dataset, without label access. The resulting method is named as ECP (Entropy scaled Conformal Prediction).

Formally, define $q_\beta(\cdot)$ as the $\beta$-th quantile of its argument, and let the base model's uncertainty measure $u_{\text{test}}$ be the "entropy quantile"

$$u_{\text{test}} = q_{1-\alpha}(\{h(x_i)\}_{i \in [N]}). \tag{3}$$

On any test covariate $x_i$, modified from Eq.(2), we scale the score function by $\max(1, u_{\text{test}})$ to form the CP prediction set

$$\mathcal{C}(x_i) := \{y \in [k] : s(x_i, y) \cdot \max(1, u_{\text{test}}) \geq \tau_D\}. \tag{4}$$

Here, we take a maximum with 1 to ensure that the prediction sets of ECP cannot be smaller than those of standard SplitCP. The pseudocode is presented as Algorithm 1 in the next subsection.

To recap, the intuition of ECP is that a larger distribution shift will result in larger entropy predicted by the base model, which then leads to a correspondingly larger up-scaling of the score function. In this way, more candidate labels have scores larger than the fixed CP threshold $\tau_D$, and the prediction set grows. Without any access to the test labels, this can help mitigate the under-coverage issue of standard SplitCP under distribution shifts, and further details are provided in our experiments (Section 5).

## 4.2 Optimizing uncertainty using TTA

While `ECP` already improves the coverage rate of `SplitCP` on several datasets, it inevitably leads to larger set sizes and, like typical post-hoc CP methods, still relies on a fixed base model. To remedy this, we refine `ECP` using *entropy minimization* (Grandvalet and Bengio, 2004; Wang et al., 2021), a classical idea in unsupervised TTA which updates the base model itself on the unlabeled test dataset. Although such techniques in unsupervised TTA have been investigated in the context of top-1 accuracy, we take a different perspective and study their ability to improve set-valued classifiers like conformal predictors.

**Key idea** Concretely, we first rewrite the entropy $h(x)$ as a loss function w.r.t. the base model's parameter $\theta$,

$$\mathcal{L}(x;\theta) := h(x) = -\sum_{y \in [k]} \pi_\theta(x)_y \log \pi_\theta(x)_y. \tag{5}$$

Our main idea is to update the base model by minimizing this loss function (or a suitable variant) on the test dataset $D_{\text{test}}$, before applying `ECP`. This brings two intuitive benefits.

- The updated base model is better suited for the shifted distribution $\mathcal{D}_{\text{test}}$, which generally improves the quality of the prediction sets built on top of it.
- The base model's entropy determines the amount of prediction set inflation due to `ECP`. By directly minimizing the entropy, the resulting prediction sets can be smaller.

**Method: `EACP`** A number of specific TTA methods have been developed to minimize the entropy, while ensuring certain notions of stability. In this work, we leverage a recent method called `ETA` (Efficient Test-time Adaptation; Niu et al. 2022), due to its simplicity and effectiveness even under continual distribution shifts (Press et al., 2023). Combining this with `ECP` results in a new CP method, which we call `EACP` (Entropy base-Adapted Conformal Prediction).

In practice, one could simply call `ETA` as a subroutine, so here we only present its high level idea for completeness. First, the test dataset $D_{\text{test}}$ is divided into a collection of batches. On each batch (i.e., $x_i$ with a collection of indices $i$), `ETA` filters the base model's outputs (i.e., softmax scores) $s(x_i, \cdot)$ by excluding the outputs similar to those already seen. Then, it reweighs the remaining indices based the associated entropy $h(x_i)$, with lower entropy (less uncertain) indices receiving higher weights. This leads to a weighted batch variant of the loss function Eq.(5), which is then minimized by performing a single gradient update. Subsequently, the updated base model is applied to `ECP` to form the prediction sets of `EACP`, according to Eq.(4).

The combined pseudo-code of `ECP` and `EACP` is provided in Algorithm 1. Here we include an *uncertainty scaling function* $f$ as a small generalization, which acts on the entropy quantile before generating the prediction sets. So far we have only considered the trivial scaling $f(x) = x$, but more choices will be studied in the next subsection.

---

**Algorithm 1** Combined pseudocode of `ECP` and `EACP`

---

**Require:** Test dataset $D_{\text{test}} = \{x_i\}_{i \in [N]}$; trained model with parameter $\theta$ and softmax score $\pi_\theta(x)_y$; targeted error rate $\alpha$; score threshold $\tau_D$ for the error rate $\alpha$, calculated on a calibration dataset $D$; uncertainty scaling function $f : \mathbb{R}_+ \to \mathbb{R}_+$.

**if** `EACP` **then**
    $\theta \leftarrow \text{ETA}(\theta, D_{\text{test}})$                        ▷ Test-time adaptation sub-routine
**end if**
$u_{\text{test}} \leftarrow q_{1-\alpha}(\{h(x_i)\}_{i \in [N]})$              ▷ Update entropy quantile, Eq.(3)
$u_{\text{test}} \leftarrow f(u_{\text{test}})$                    ▷ Modify the entropy adjustment factor
**for** $x_i \in D_{\text{test}}$ **do**
    **return** $\mathcal{C}(x_i) := \{y \in [k] : s(x_i, y) \cdot \max(1, u_{\text{test}}) \geq \tau_D\}$      ▷ Predict the label set, Eq.(4)
**end for**

---

In Section 5, we demonstrate that `EACP` can further improve the empirical performance of `ECP`, by increasing the coverage rate while maintaining informative set sizes.

### 4.3 UNCERTAINTY SCALING FUNCTION

In Eq.(4), we essentially scale the score functions linearly by the entropy quantile $u_{\text{test}}$ of the base model. However, this can be adjusted more generally by any (potentially non-linear) function $f(\cdot)$. The best choice of $f(\cdot)$ depends on the unknown relation between $u_{\text{test}}$ and the $(1 - \alpha)$-quantile of the ground truth labels' conformal scores, denoted as[5]

$$\tau_{\text{test}} := q_{1-\alpha}\left[s(x_i, y_i); x_i \in D_{\text{test}}\right].$$

Specifically, such an optimal $f(\cdot)$ should satisfy $f(u_{\text{test}}) = \tau_D/\tau_{\text{test}}$.

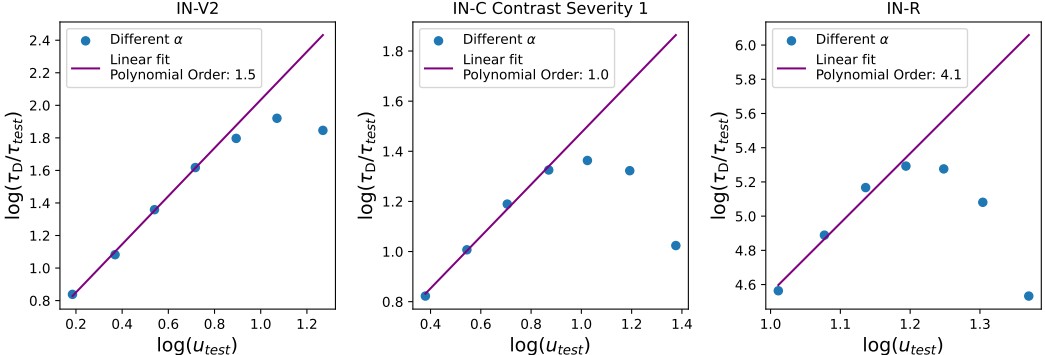

Figure 3: $u_{\text{test}}$ versus $\tau_D/\tau_{\text{test}}$ on a log-log scale. On mild and moderate distribution shifts, the linear fit on the log-log plot has slope between 1 and 2. This suggests the effectiveness of using a linear or quadratic function as $f(\cdot)$, which acts on the entropy quantile. However, we also observe that a higher-order polynomial is required on more difficult shifts, such as ImageNet-R.

While finding this optimal $f(\cdot)$ is obviously infeasible without observing the ground truth labels at test time, in Figure 3 we empirically evaluate the ideal choice in a post-hoc manner, across different datasets, in order to demonstrate the insights. Recall that both $u_{\text{test}}$ and $\tau_{\text{test}}$ depend on the desired error rate $\alpha$. Therefore, for each dataset, we vary $\alpha$ and plot the resulting $u_{\text{test}}$ versus $\tau_D/\tau_{\text{test}}$ on a log-log scale. If we mildly restrict $f(\cdot)$ to the family of polynomials, then its optimal order can be approximated by the slope of a linear fit on the log-log plot. We only use not-extremely-small $\alpha$ values (i.e., the lower left corner on the plot) for the linear fit, since it is closer to the typical practice and less prone to noise.

Figure 3 shows that the optimal polynomial order generally increases with the severity of the distribution shift, which is consistent with the fact that a larger polynomial order would lead to larger prediction sets using our methods. While end-users can refine $f(\cdot)$ based on a preference towards ensuring coverage or small set sizes, we will empirically validate that our methods with either linear scaling (denoted by $\texttt{ECP}_1$ / $\texttt{EACP}_1$) or quadratic scaling (denoted by $\texttt{ECP}_2$ / $\texttt{EACP}_2$) perform well in a wide range of settings.

## 5 EXPERIMENTS

We conduct experiments across a number of large-scale datasets and neural network architectures. Our setup builds on the standard $\texttt{SplitCP}$ procedure introduced in Section 3, which relies on a held-out, in-distribution, "development set" for calibrating the CP threshold. On ImageNet variants, we split the original ImageNet development set (i.e., not used for model training) into a CP calibration set consisting of 25,000 samples, and an in-distribution test set (sometimes called the validation set in the CP literature). The readers are referred to Appendix B for more details.

The conformal threshold is found on the calibration set, and used in subsequent distribution-shifted settings. Importantly, *after the conformal threshold is estimated in-distribution, all subsequent steps are unsupervised.* We show results on both stationary and continuously shifting test distributions.

---

[5]Recall that the data we face at test time is denoted as $\{x_i, y_i\}_{i\in[N]}$, with the observed part (covariates) denoted as $D_{\text{test}} = \{x_i\}_{i\in[N]}$.

**Baselines**   We compare our proposed methods to the following baselines:

- `NAIVE`: generating prediction sets by including classes until their cumulative softmax score is greater or equal to $1 - \alpha$ (the target coverage level). This is generated by the base model itself, without the CP post-processing.
- Standard `SplitCP`: applying the CP threshold directly on the distribution-shifted data.
- `SplitCP` with `ETA`: applying the CP threshold while updating the base model using `ETA`.

Furthermore, in settings with stationary distribution shifts, we compare to Robust Conformal (`RC`; Cauchois et al. 2024), an existing CP algorithm that handles distribution shifts via robust optimization. In settings with continual distribution shifts, we compare to a number of OCO-based algorithms (Bhatnagar et al., 2023a; Gibbs and Candès, 2024; Zhang et al., 2024) that require additional access to the ground truth labels.

In all experiments, the target coverage rate is set to 0.90. We also analyze our methods with both linear and quadratic scaling, as described in Section 4.3.

**Datasets**   We investigate a number of ImageNet (Deng et al., 2009) variants including: ImageNet-V2 (Recht et al., 2019), ImageNet-R (Hendrycks et al., 2021a), and ImageNet-A (Hendrycks et al., 2021b). We also test our approach on datasets from the WILDS Benchmark (Koh et al., 2021) which represent in-the-wild distribution shifts across many real world applications, including iWildCam (animal trap images), RXRX1 (cellular images), and FMOW (satellite images).

While the previous datasets present a single distribution shift, the ImageNet-C (Hendrycks and Dietterich, 2019) dataset allows us to investigate shifts across many types and severities. Specifically, ImageNet-C applies 19 visual corruptions to the ImageNet validation set across four corruption categories — noise, blur, weather, and digital, with five severity levels for each corruption. See Appendix B.1 for more details on the datasets.

## 5.1 STATIONARY SHIFTS

Table 1: ECP and EACP can achieve very competitive empirical coverage rates on a number of distribution-shifted datasets, across a variety of imaging domains (ecological, cellular, satellite, etc). All results are from ResNet-50 models except FMOW, which uses a DenseNet-121 (Huang et al., 2016). Quadratic uncertainty scaling provides better coverage rates, however, linear scaling results in smaller set sizes.

|  | Method | ImageNet-V2 | ImageNet-R | ImageNet-A | iWildCam | RXRX1 | FMOW |
|---|---|---|---|---|---|---|---|
|  | SplitCP | 0.81 | 0.50 | 0.03 | 0.84 | 0.84 | 0.87 |
| Coverage | NAIVE | 0.88 | 0.69 | 0.14 | 0.76 | 0.48 | 0.83 |
|  | RC | 0.88[6] | 0.63 | 0.14 | 0.99 | 0.91 | 0.93 |
|  | ETA | 0.81 | 0.62 | 0.05 | 0.84 | 0.87 | 0.87 |
|  | $ECP_1$ | 0.86 | 0.61 | 0.10 | 0.84 | 0.87 | 0.93 |
|  | $ECP_2$ | **0.91** | 0.72 | 0.27 | 0.88 | 0.90 | **0.96** |
|  | $EACP_1$ | 0.86 | 0.71 | 0.14 | 0.84 | 0.90 | 0.93 |
|  | $EACP_2$ | **0.91** | **0.80** | **0.30** | **0.89** | **0.93** | 0.94 |
|  | SplitCP | 2.5 | 3.4 | 3.4 | 3.9 | 81.8 | 6.2 |
| Set Size | NAIVE | 11.7 | 20.9 | 12.7 | 2.5 | 6.4 | 5.8 |
|  | RC | 5.5 | 10.7 | 9.6 | 125 | 166 | 10.2 |
|  | ETA | 2.5 | 3.0 | 3.6 | 3.8 | 100 | 6.5 |
|  | $ECP_1$ | 4.2 | 9.1 | 7.4 | 3.8 | 105 | 10.3 |
|  | $ECP_2$ | 7.6 | 23.3 | 15.1 | 5.5 | 137 | 15.3 |
|  | $EACP_1$ | 4.5 | 6.8 | 8.7 | 3.7 | 133 | 11.1 |
|  | $EACP_2$ | 8.7 | 16.1 | 10.1 | 5.6 | 177 | 16.4 |

---

[6]Note this result slightly differs from that reported in (Cauchois et al., 2024), as they evaluate on an alternate variant of ImageNet-V2.

Table 1 summarizes our results on various natural distribution-shifted datasets. We observe that `SplitCP` (with or without TTA) can exhibit significant gaps with respect to the target coverage rate, whereas `ECP` closes the gap quite effectively while maintaining meaningful set sizes. Coverage is further improved via `EACP`, which also helps reducing set sizes on some datasets. In general, we also observe an improvement over `RC` and `NAIVE`: the linear scaling variant of our methods has similar coverage rates as these baselines, while the set sizes are typically smaller.

Here we can see the trade-off between linear and quadratic uncertainty scaling. $\text{EACP}_2$ consistently achieves higher coverage rates, however this also leads to "over-coverage" on some datasets and thus larger sets. In contrast, $\text{EACP}_1$ leads to lower coverage but also smaller set sizes. This trade-off can be selected by end-users depending on their preference for more accurate or more efficient prediction sets. In subsequent experiments, we will focus on demonstrating the benefit of $\text{EACP}_2$ on coverage, while noting that the observed set sizes are nonetheless practically useful and far from trivial.

Table 2: Coverage on four different corruption types representing each ImageNet-C category. Compared to the baselines, $\text{ECP}_2$ closes the coverage gap on most severity levels, while $\text{EACP}_2$ further improves this by achieving the target coverage rate 0.90 on nearly all corruption types and severities.

| Method | Contrast | | | | | Brightness | | | | | Gaussian Noise | | | | | Motion Blur | | | | |
|---|---|---|---|---|---|---|---|---|---|---|---|---|---|---|---|---|---|---|---|---|
| | 1 | 2 | 3 | 4 | 5 | 1 | 2 | 3 | 4 | 5 | 1 | 2 | 3 | 4 | 5 | 1 | 2 | 3 | 4 | 5 |
| NAIVE | 0.91 | 0.89 | 0.87 | 0.83 | 0.76 | 0.92 | 0.92 | 0.91 | 0.91 | 0.90 | 0.88 | 0.85 | 0.79 | 0.69 | 0.79 | 0.91 | 0.90 | 0.85 | 0.77 | 0.71 |
| SplitCP (Sadinle et al., 2019) | 0.83 | 0.78 | 0.66 | 0.36 | 0.09 | 0.88 | 0.87 | 0.86 | 0.83 | 0.78 | 0.79 | 0.69 | 0.50 | 0.26 | 0.07 | 0.83 | 0.74 | 0.57 | 0.37 | 0.27 |
| ETA (Niu et al., 2022) | 0.87 | 0.86 | 0.84 | 0.79 | 0.63 | 0.88 | 0.88 | 0.87 | 0.86 | 0.84 | 0.86 | 0.82 | 0.76 | 0.69 | 0.54 | 0.86 | 0.84 | 0.80 | 0.73 | 0.68 |
| $\text{ECP}_2$ (ours) | 0.93 | 0.92 | 0.89 | 0.79 | 0.60 | 0.94 | 0.94 | 0.94 | 0.93 | 0.92 | 0.92 | 0.88 | 0.80 | 0.86 | 0.38 | 0.94 | 0.92 | 0.86 | 0.75 | 0.68 |
| $\text{EACP}_2$ (ours) | 0.93 | 0.93 | 0.93 | 0.92 | 0.87 | 0.93 | 0.93 | 0.93 | 0.93 | 0.93 | 0.93 | 0.93 | 0.92 | 0.90 | 0.84 | 0.93 | 0.92 | 0.92 | 0.91 | 0.89 |

In Table 2, we show fine-grained results on one corruption type for each ImageNet-C category, and across each severity level. Here, we can see the benefit of leveraging an uncertainty notion that can be directly minimized and refined on new test samples. Specifically, $\text{EACP}_2$ is able to recover the target coverage rate on almost all corruption types and severities.

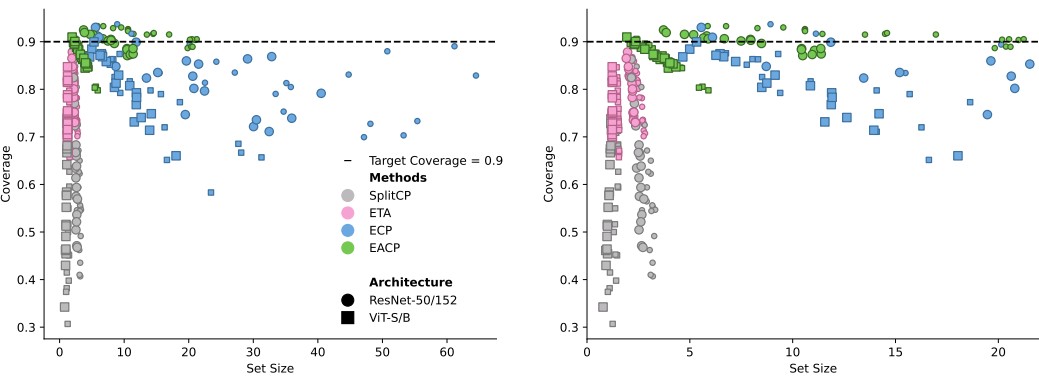

Figure 4: Our $\text{EACP}_2$ method is able to improve coverage using various neural network models and architectures, under a diverse range of distribution shifts. It consistently "hugs" the desired coverage rate, while maintaining practical set sizes. Results are averaged across five severity levels for each corruption type in the ImageNet-C dataset. We zoom in on the **right** to clearly see the benefit of adapting at test time on coverage and set sizes. Larger markers reflect a larger neural network parameter count.

Next, Figure 4 contains the results using neural networks of various architectures and parameter counts, on all 19 corruption types of ImageNet-C (average across five severity levels). Besides showing the superior performance of our methods, we observe that the `SplitCP` baseline (with and without TTA) generates prediction sets with little variance in the set sizes, regardless of the achieved coverage rates. We argue that this is an undesirable behavior, as the set sizes themselves are often used to encode uncertainty evaluations by set-valued classifiers. Our results demonstrate that explicitly incorporating the base model's own uncertainty into CP can help mitigating this issue.

## 5.2 CONTINUOUS SHIFTS

Finally, we investigate continuous distribution shifts, and the results are shown in Table 3. This has been previously studied under *online conformal prediction*, and we build on the experimental setup of (Bhatnagar et al., 2023b; Zhang et al., 2024). Specifically, the environment shifts between ImageNet-C severity level 1 to level 5 (either suddenly or gradually), while sampling random corruptions at each corresponding severity. We emphasize that this is a particularly challenging task, as it presents a continuous shift in both the magnitude as well as type of corruption. See Appendix B.2 for more details on this experiment. Here, we compare with existing supervised methods that rely on the correct label being revealed after every prediction.

Table 3: We evaluate performance on the challenging setting of continuously shifting distributions. The "label free" column denotes whether a method relies on labels at test-time from the target data. We recall that SplitCP does not adapt to new data. In addition to the **average coverage** (↑) and **average size** (↓), we also measure the worst local corruption error $\mathbf{LCE_{128}}$ (↓) and worst local set size $\mathbf{LSS_{128}}$, (↓) on a sliding window of 128 test points.

| | | Gradual shift | | | | Sudden shift | | | |
|---|---|---|---|---|---|---|---|---|---|
| Label Free | Method | Avg. Cov | Avg. Size | $LCE_{128}$ | $LSS_{128}$ | Avg. Cov | Avg. Size | $LCE_{128}$ | $LSS_{128}$ |
| - | SplitCP (Sadinle et al., 2019) | 0.59 | 3.1 | 0.70 | 3.6 | 0.59 | 2.8 | 0.71 | 3.5 |
| ✗ | SAOCP (Bhatnagar et al., 2023a) | 0.79 | 145 | 0.24 | 353 | 0.78 | 139 | 0.28 | 349 |
| ✗ | DtACI (Gibbs and Candès, 2024) | 0.90 | 101 | 0.07 | 455 | 0.90 | 142 | 0.09 | 450 |
| ✗ | MAGL-D (Zhang et al., 2024) | 0.90 | 403 | 0.05 | 856 | 0.90 | 355 | 0.05 | 844 |
| ✗ | MAGL (Zhang et al., 2024) | 0.90 | 117 | 0.06 | 573 | 0.90 | 168 | 0.3 | 704 |
| ✗ | MAGDIS (Zhang et al., 2024) | 0.90 | 417 | 0.06 | 841 | 0.90 | 372 | 0.07 | 852 |
| ✓ | ETA (Niu et al., 2022) | 0.69 | 2.9 | 0.52 | 3.4 | 0.67 | 2.7 | 0.54 | 3.5 |
| ✓ | $ECP_2$ (ours) | 0.84 | 36.6 | 0.35 | 90.4 | 0.82 | 37.5 | 0.38 | 88.5 |
| ✓ | $EACP_2$ (ours) | 0.88 | 22.4 | 0.20 | 47.8 | 0.86 | 23.1 | 0.28 | 55.7 |

Overall, our methods demonstrate competitive performance with respect to supervised baselines: the average set sizes are significantly smaller despite a slight drop in the average coverage rate. In addition, we also measure the local coverage error $LCE_{128}$ across the worst sliding window of 128 samples, and similarly the worst local set size, $LSS_{128}$. While the supervised methods unsurprisingly result in better local coverage, they also lead to local set sizes that are much larger.

## 6 CONCLUSION

This paper studies how to improve set-valued classification methods on distribution-shifted data, without relying on labels from the target dataset. This is an important challenge in many real world settings, where exchangeability assumptions are violated and labels may be difficult to attain. We propose an uncertainty-aware method based on the prediction entropy (ECP), and leverage unsupervised test time adaptation to update the base model and refine its uncertainty (EACP). We demonstrate that the proposed methods are able to recover the desired error rate on a wide range of distribution shifts, while maintaining efficient set sizes. Furthermore, they are even competitive with supervised approaches on challenging and continuously shifting distributions. We hope this inspires future works continuing to tackle this important challenge.

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

APPENDIX

In Appendix A, we provide an extensive overview of additional related works. Further, Appendix B contains detailed information on our studied datasets and experimental protocols, including TTA hyper-parameters, CP procedure, and the setup for continual distribution shifts. Appendix C discusses other possible uncertainty measures and their deficiencies. Appendix D contains additional experiment results.

## A  ADDITIONAL RELATED WORKS

**CP in decision making**  Our interest in the considered setting – distribution shifts without test time labels – is mainly motivated by the growing applications of CP in autonomous decision making. A very much incomplete list: see (Lekeufack et al., 2023) for a generic treatment; (Lindemann et al., 2023) for trajectory optimization in robotics; (Yang and Pavone, 2023; Gao et al., 2024) for 3D vision; (Kumar et al., 2023; Cherian et al., 2024; Gui et al., 2024; Mohri and Hashimoto, 2024; Quach et al., 2024) for large language models (LLMs); and (Ren et al., 2023) for LLM-powered robotics.

**CP under distribution shifts**  As discussed in the main paper, considerable efforts have been devoted to developing CP methods robust to distribution shifts. We now survey two possible directions and their respective limitations.

- The first direction does not require test time labels, but the distribution shift is assumed to be simple in some sense. For example, Tibshirani et al. (2019) studied CP under *covariate shifts*, where the distribution of the label $y$ conditioned on the covariate $x$ remains unchanged. Here, it suffices to use the classical *likelihood ratio reweighting* on the calibration dataset, but accurately estimating the likelihood ratio can be challenging in practice. Another idea is to take a robust optimization perspective by assuming a certain maximum level of distribution shift and protecting against the worst case, e.g., (Roth, 2022, Chapter 8) and (Cauchois et al., 2024). The weakness here is the sensitivity to the hyperparameter, and the obtained prediction sets could be overly conservative.

  Various works built on these two ideas. Barber et al. (2023) generalized the reweighting idea to handle mild but general distribution shifts, but choosing the weights is generally unclear in practice. Ai and Ren (2024) tackled general distribution shifts by combining reweighting and robust optimization, which also combines the strengths and limitations from the two sides. Ge et al. (2024) extended the two ideas to the aggregation of multiple CP algorithms.

- The second direction is connecting CP to adversarial online learning. A line of works (Gibbs and Candes, 2021; Angelopoulos et al., 2023; Gibbs and Candès, 2023; Bhatnagar et al., 2023b; Zhang et al., 2024) applied regret minimization algorithms in OCO to select the score threshold in CP, and Bastani et al. (2022) achieved this task using *multicalibration*. By relaxing the CP objective from the *coverage probability* to the *post-hoc coverage frequency*, these methods can handle arbitrary continual distribution shifts. However, they require the true label to be provided after every prediction, which is a limiting requirement for many use cases in autonomous decision making. Our experiments will show that it is possible to achieve comparable performance in these settings without this limitation, i.e., being "label free".

## B  EXPERIMENTAL DETAILS

### B.1  DATASET DETAILS

We perform experiments on a number of large-scale datasets that are frequently used to evaluate deep learning performance under distribution shift (Koh et al., 2021; Wang et al., 2021; Minderer et al., 2021; Niu et al., 2022; Zhang et al., 2022; Bhatnagar et al., 2023a; Zhang et al., 2024):

- **ImageNet-V2** (Recht et al., 2019) is an ImageNet test-set that contains 10,000 images that were collected by closely following the original ImageNet data collection process.
- **ImageNet-R** (Hendrycks et al., 2021a) includes renditions (e.g., paintings, sculptures, drawings, etc.) of 200 ImageNet classes, resulting in a test set of 30,000 images.
- **ImageNet-A** (Hendrycks et al., 2021b) consists of 7,500 real-world, unmodified, and naturally occurring adversarial images which a ResNet-50 model failed to correctly classify.

- **ImageNet-C** (Hendrycks and Dietterich, 2019) applies 19 visual corruptions across four categories and at five severity levels to the original ImageNet validation set.
- **iWildCam** (Koh et al., 2021; Beery et al., 2020) contains camera-trap images from different areas of the world, representing geographic distribution-shift. It includes a validation set of 7,314 images from the same camera traps the model was trained on, which is used as our calibration data, as well as 42,791 images from different camera traps that is used as our test set. The images contain one of the 182 possible animal species.
- **RXRX1** (Koh et al., 2021; Taylor et al., 2019) consists of high resolution fluorescent microscopy images of human cells which have been given one of 1,139 genetic treatments, with the goal of generalizing across experimental batches. It is split into a 40,612 in-distribution validation set and 34,432 test set.
- **FMOW** (Koh et al., 2021; Christie et al., 2018) is a satellite imaging dataset with the goal of classifying images into one of 62 different land use or building types. It consists of 11,483 validation images from the years from 2002–2013, and 22,108 test images from the years from 2016–2018.

### B.2 EXPERIMENTAL PROTOCOLS

**Conformal prediction**   Our split conformal prediction set-up follows previous works (Angelopoulos et al., 2021; Angelopoulos and Bates, 2023), which divides a held-out dataset into a calibration and test set. On ImageNet variants, we split the original validation set in half to produce 25,000 calibration points and 25,000 in-distribution test points. The calibrated scores and / or threshold are then used for subsequent distribution-shifted data. On the WILDS datasets, we similarly split the in-distribution validation sets.

**Adaptation procedure**   Our ImageNet-based experiments are conducted on pre-trained ResNets provided by the *torchvision* library[7], and ViTs provided by the *timm* library [8]. Experiments on WILDS datasets are conducted using pre-trained models provided by the authors of that study [9]. For `EACP` and `ETA`, we closely follow the optimization hyperparameters from the original paper (Niu et al., 2022): we use SGD optimizer with a momentum of 0.9 and learning rate of 0.00025. We use a batch size of 64 for all ImageNet experiments, 128 for RXRX1 and FMOW, and 42 for iWildCam. Our experiments are conducted using a single NVIDIA A40 GPU.

**Continuous shift**   We adopt a slightly modified version of the experimental design for continuous distribution shift presented in previous works (Bhatnagar et al., 2023a; Zhang et al., 2024). This involves sampling random corruptions from the ImageNet-C dataset under two regimes: **gradual shifts** where the severity level first increases in order from $\{1, ..., 5\}$ then decreases from $\{5, ..., 1\}$, and sudden shifts where the severity level alternates between 1 and 5. In addition to sampling random corruptions, we also consider in Figure 6 results on the "easier" setting of shifting severities on a single corruption type.

## C   WHAT IS THE RIGHT MEASURE OF UNCERTAINTY?

Although in Section 4.1 we propose adjusting the conformal scores by the prediction entropy of the base model, it is worth asking if there exist other notions of uncertainty that may instead be used. Here, we consider two additional uncertainty measures and their relation with the softmax value of the true label (which is ultimately what we would like to include in our prediction set), and show they are ill-suited for our task. Firstly, in Figure 5a we consider the variance of the softmax scores. Perhaps surprisingly, we see that distribution shift most often leads to a *smaller* variance, thus conveying that the base model is *less* uncertain. This suggests that softmax variance is a deficient uncertainty measure as it fails to capture the actual underlying uncertainty on distribution-shifted data.

We also consider $1-$ maximum softmax score as another possible uncertainty measure, and see in Figure 5b that distribution shift is associated with smaller maximum softmax values. Unlike softmax

---

[7]https://github.com/pytorch/vision
[8]https://github.com/huggingface/pytorch-image-models
[9]https://github.com/p-lambda/wilds

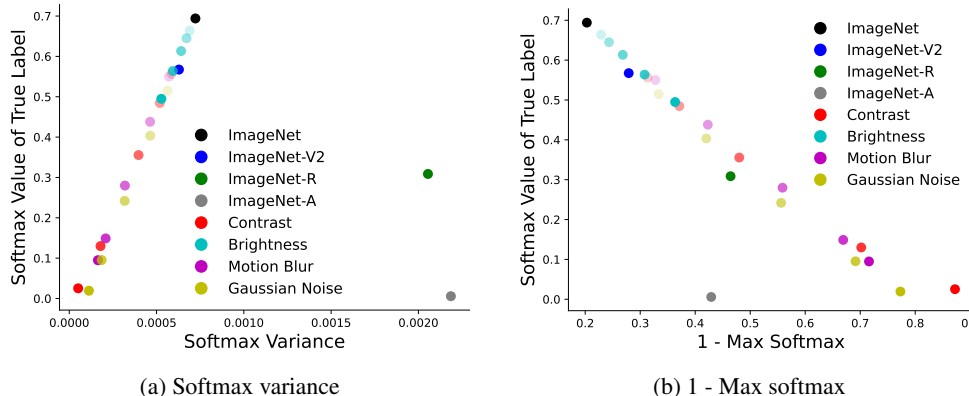

(a) Softmax variance          (b) 1 - Max softmax

Figure 5: Similarly to Figure 1, we present the relation between different uncertainty measures and the average score of the true label. We see that softmax variance (left) has an inverse relation with distribution shift, and $1-$ maximum softmax is a bounded metric that may provide an insufficient adjustment.

Table 4: Our proposed `EACP` performs well with other TTA methods, as seen here using Tent Wang et al. (2021) as the TTA update.

| Dataset | SplitCP (coverage / set size) | Tent (coverage / set size) | $\text{ECP}_2$ (coverage / set size) | $\text{EACP}_2$ (coverage / set size) |
|---|---|---|---|---|
| ImageNet-V2 | 0.81 / 2.5 | 0.81 / 2.6 | 0.91 / 8.0 | 0.92 / 9.6 |
| ImageNet-R | 0.50 / 3.2 | 0.58 / 3.3 | 0.73 / 23 | 0.77 / 17 |
| ImageNet-A | 0.07 / 1.5 | 0.21 / 3.1 | 0.58 / 204 | 0.40 / 24 |
| iWildCam | 0.83 / 3.5 | 0.81 / 2.6 | 0.89 / 5.7 | 0.85 / 3.4 |
| RXRX1 | 0.85 / 83 | 0.87 / 101 | 0.90 / 136.7 | 0.92 / 176 |
| FMOW | 0.87 / 6.3 | 0.85 / 5.7 | 0.96 / 15.6 | 0.94 / 13.4 |

variance, this does appear to better capture the uncertainty, as we would expect the base model to be less confident on distribution-shifted data. However, this uncertainty measure can still only take a maximum value of one and thus may not provide necessary adjustment magnitude, and it is unknown if it can reliably update the base model label-free.

While there may exist better uncertainty measures that future works can explore, these results suggest that the prediction entropy is a simple and reliable measure for conformal adjustments that can effectively capture the underlying uncertainty.

## D ADDITIONAL EXPERIMENTS

### D.1 OTHER TTA METHODS

We investigate our methods performance with another base TTA method in Table 4. Here, we use the `Tent` update (Wang et al., 2021), which is a simpler version of `ETA` with no re-weighing of the entropy loss. While our proposed methods are also compatible with `Tent`, we notice that the more powerful `ETA` leads to better coverage and set sizes as seen in Table 1. We can expect that additional improvements in TTA will similarly lead to improvements in our `EACP` method.

### D.2 MORE ARCHITECTURE COMPARISONS

In Table 5, we further demonstrate our methods improvements to coverage loss on natural distribution shifts using diverse neural network architectures. As expected, larger and more accurate neural networks result in better coverage and smaller set sizes using `ECP` and `EACP` This is encouraging as it demonstrates our methods can scale along with the underlying model.

Table 5: On natural distribution shifts, the performance of our methods scale well with the performance of the base classifier. This is encouraging as it suggests compatibility

| Dataset | Model | SplitCP (coverage / set size) | ETA (coverage / set size) | ECP$_2$ (coverage / set size) | EACP$_2$ (coverage / set size) |
|---|---|---|---|---|---|
| **ImageNet-V2** | Resnet50 | 0.81 / 2.5 | 0.81 / 2.5 | 0.91 / 7.6 | 0.91 / 8.7 |
| | Resnet152 | 0.81 / 2.0 | 0.81 / 2.1 | 0.89 / 4.6 | 0.91 / 6.3 |
| | Vit-S | 0.80 / 1.5 | 0.80 / 1.5 | 0.90 / 3.4 | 0.90 / 3.4 |
| | ViT-B | 0.80 / 1.2 | 0.80 / 1.2 | 0.90 / 2.4 | 0.90 / 2.4 |
| **ImageNet-R** | Resnet50 | 0.50 / 3.4 | 0.62 / 3.0 | 0.72 / 23.3 | 0.80 / 16.1 |
| | Resnet152 | 0.53 / 2.7 | 0.60 / 2.6 | 0.71 / 15.3 | 0.79 / 17.3 |
| | Vit-S | 0.52 / 1.3 | 0.53 / 1.3 | 0.74 / 12.3 | 0.75 / 11.8 |
| | ViT-B | 0.58 / 0.9 | 0.59 / 0.9 | 0.78 / 8.3 | 0.79 / 8.0 |
| **ImageNet-A** | Resnet50 | 0.03 / 3.4 | 0.05 / 3.6 | 0.27 / 15.1 | 0.30 / 19.1 |
| | Resnet152 | 0.18 / 3.0 | 0.17 / 3.3 | 0.43 / 11.8 | 0.50 / 19.6 |
| | Vit-S | 0.37 / 1.7 | 0.37 / 1.7 | 0.65 / 8.4 | 0.66 / 8.3 |
| | ViT-B | 0.47 / 1.2 | 0.47 / 1.2 | 0.76 / 6.5 | 0.76 / 6.4 |

## D.3 CONTINUOUS SHIFTS

In Figure 6, we visualize the coverage and set-sizes of our unsupervised methods and a number of supervised baselines on the previously described continuous distribution shifts. We show results on random corruption types as well as fixed corruption types. Our proposed methods perform well across all these settings; they closely maintain coverage even on sudden and severe shifts, while leading to substantially smaller set sizes than the baselines.

## D.4 IMAGENET-C ALL SEVERITY LEVELS

In Figure 7, we present full results across all ImageNet-C severity levels. We see that our method is effective in recovering coverage even under many highly severe distribution shifts, and nearly always recovers the desired coverage on less severe shifts.

## D.5 ORACLE RESULTS

Here we compare our methods with an oracle that has observed labels from the distribution-shifted dataset. Specifically, the oracle is the THR conformal prediction method (Sadinle et al., 2019) that has been calibrated on half of the distribution-shifted dataset, following regular split conformal. Since the oracle is guaranteed to provide the desired coverage level in the set-up, our comparison focuses on the prediction set sizes; we refer to the main paper for coverage comparisons. We observe in Table 6 that in every case except FMOW, a variant of ECP and EACP achieves smaller set sizes than the oracle. In Table 7, EACP consistently achieves substantially smaller set sizes on ImageNet-C while also recovering error targets (see Table 2). We reiterate here that smaller sets are preferred if error rates are maintained.

Table 6: ECP and EACP achieve prediction set sizes that are often equal or smaller than the oracle method. Coverage rate is 0.90.

| | Method | ImageNet-V2 | ImageNet-R | ImageNet-A | iWildCam | RXRX1 | FMOW |
|---|---|---|---|---|---|---|---|
| | ORACLE | 6.8 | 79.0 | 95.3 | 6.6 | 140 | 7.87 |
| Set Size | ECP$_1$ | 4.2 | 9.1 | 7.4 | 3.8 | 105 | 10.3 |
| | ECP$_2$ | 7.6 | 23.3 | 15.1 | 5.5 | 137 | 15.3 |
| | EACP$_1$ | 4.5 | 6.8 | 8.7 | 3.7 | 133 | 11.1 |
| | EACP$_2$ | 8.7 | 16.1 | 10.1 | 5.6 | 177 | 16.4 |

## D.6 IN-DISTRIBUTION RESULTS

In Table 8, we observe that our methods maintain coverage and reasonable set sizes on in-distribution data.

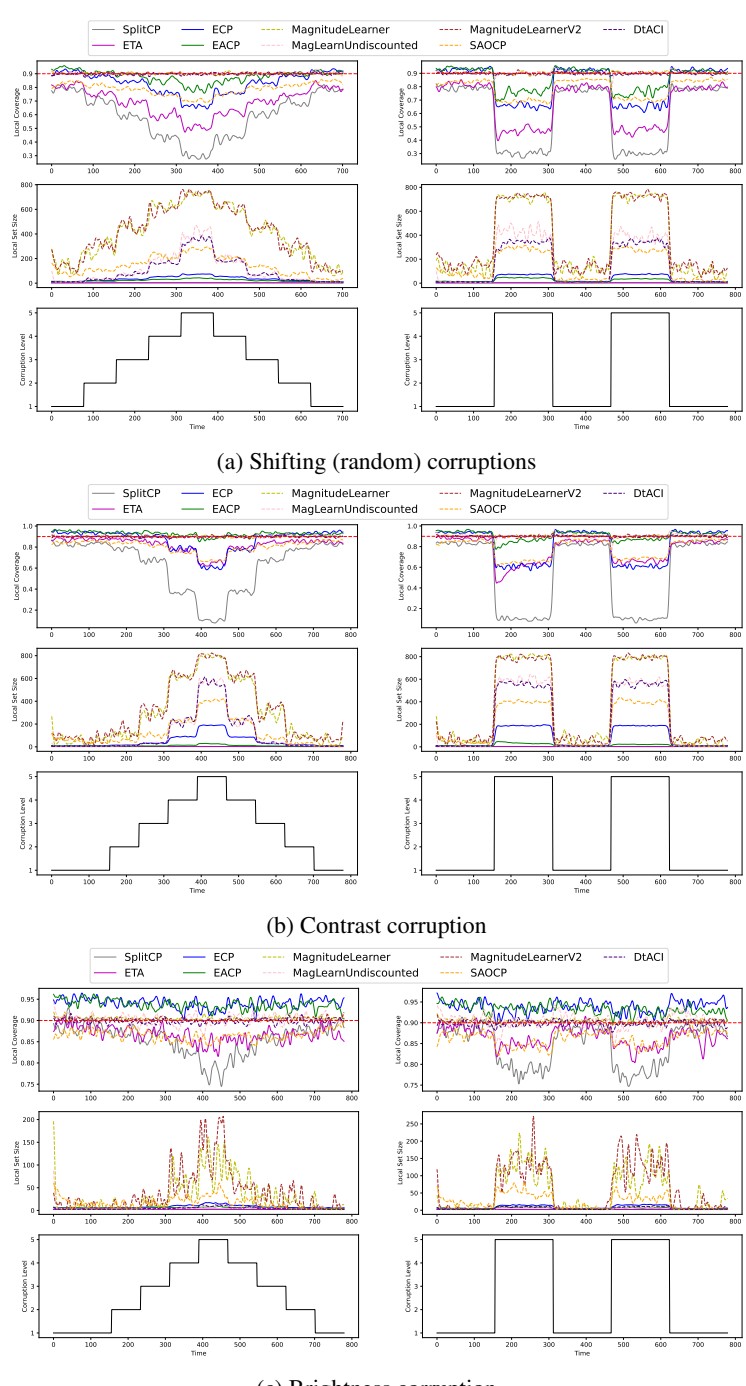

(a) Shifting (random) corruptions

(b) Contrast corruption

(c) Brightness corruption

## D.7 COMPARISON WITH WEIGHTED CP

Tibshirani et al. (2019) present a method for improving coverage under covariate shift by re-weighing calibration scores based on an estimated likelihood ratio (wcp). Although estimating likelihood ratios in our setting is challenging, we nonetheless present a comparison here for completeness. We follow their approach and train a probabilistic classifier, here a CNN, on each calibration-test pair.

Table 9 suggests that this method may have limited performance in our studied setting. This may be due to the challenge in estimating accurate likelihood ratios in high-dimensional settings, (Cauchois et al., 2024). We do not claim that wcp definitely *cannot* perform well, however the sparsity of

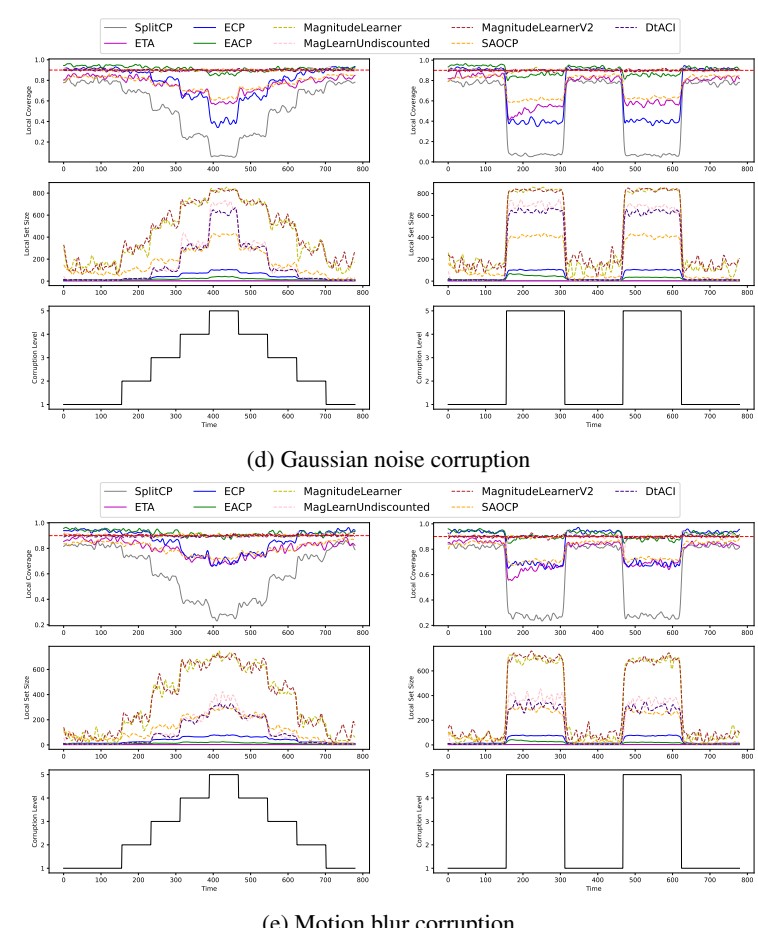

(d) Gaussian noise corruption

(e) Motion blur corruption

Figure 6: Our unsupervised methods ECP and EACP are able to provide nearly the same empirical coverage, and considerably smaller set sizes, that supervised methods on continuously shifting distributions. Dashed lines denote methods that rely on a ground truth label being revealed at test time.

Table 7: Comparison of ECP and EACP on a subset of synthetic shifts. The numbers refer to severity level.

|  | Method | Contrast | | | Brightness | | | Gaussian Noise | | | Motion Blur | | |
|---|---|---|---|---|---|---|---|---|---|---|---|---|---|
|  |  | 1 | 3 | 5 | 1 | 3 | 5 | 1 | 3 | 5 | 1 | 3 | 5 |
|  | ORACLE | 5.5 | 30.3 | 562 | 2.5 | 3.7 | 9.8 | 6.2 | 70.6 | 317 | 9.7 | 101 | 638 |
| Set Size | $\text{ECP}_2$ | 10.5 | 27.8 | 180 | 5.3 | 7.7 | 14.9 | 10.0 | 43.1 | 109 | 12.9 | 43.7 | 79.0 |
|  | $\text{EACP}_2$ | 5.5 | 7.4 | 25 | 4.5 | 5.7 | 7.6 | 5.7 | 16.0 | 42.7 | 6.3 | 12.8 | 25.5 |

previous literature here suggests that further studies may be required. Finally, note that wcp is ill-suited for the case of continuously shifting distributions, further limiting its general applicability.

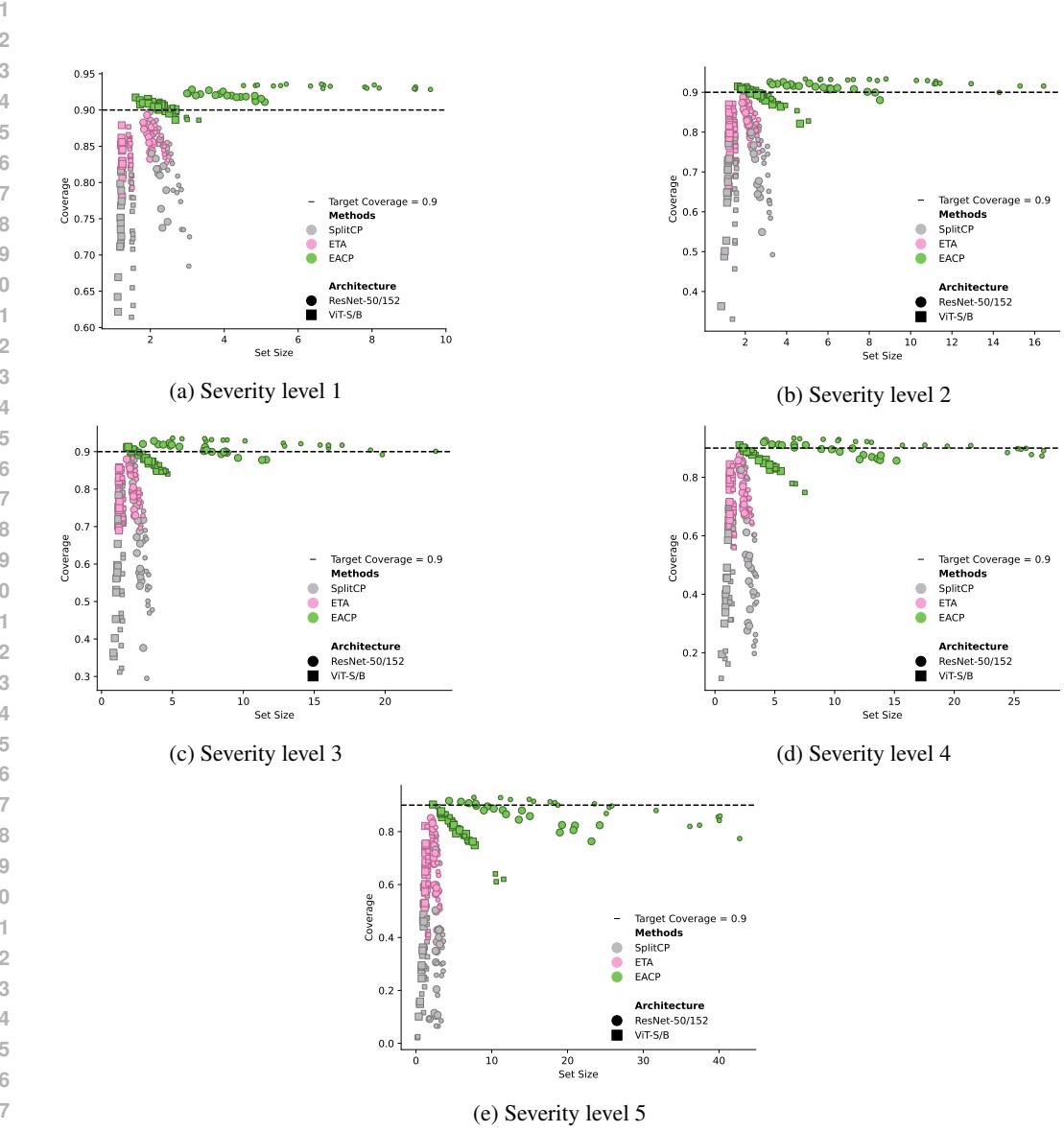

(a) Severity level 1

(b) Severity level 2

(c) Severity level 3

(d) Severity level 4

(e) Severity level 5

Figure 7: Performance on 19 ImageNet-C corruptions on each severity level. $EACP_2$ hugs the desired coverage line on nearly all severity levels. Larger markers indicate larger parameter count.

Table 8: Results on in-distribution data using ImageNet-1k validation set.

| | SplitCP | $ECP_1$ | $ECP_2$ | $EACP_1$ | $EACP_2$ |
|---|---|---|---|---|---|
| Coverage | 0.90 | 0.92 | 0.94 | 0.91 | 0.93 |
| Set size | 2.1 | 2.8 | 4.2 | 2.8 | 4.1 |

Table 9: The `wcp` method appears to provide minimal coverage improvements in this setting, possibly due to the difficulty in estimating likelihood ratios.

| | Method | ImageNet-V2 | ImageNet-R | ImageNet-A |
|---|---|---|---|---|
| | SplitCP | 0.81 | 0.50 | 0.03 |
| | wcp | 0.82 | 0.35 | 0.06 |
| Coverage | $ECP_2$ | 0.91 | 0.72 | 0.27 |
| | $EACP_2$ | 0.91 | 0.80 | 0.30 |
| | SplitCP | 2.5 | 3.4 | 3.4 |
| | wcp | 2.6 | 0.74 | 4.3 |
| Set Size | $ECP_2$ | 7.6 | 23.3 | 15.1 |
| | $EACP_2$ | 8.7 | 16.1 | 10.1 |

