# OpenReview forum: "Adapting Prediction Sets to Distribution Shifts Without Labels"
_ICLR.cc/2025/Conference — Submitted to ICLR 2025_

### Official Review · Reviewer_cR3g · 2024-10-18

**Soundness:** 3
**Presentation:** 3
**Contribution:** 2
**Rating:** 5
**Confidence:** 3

**Summary:**

Conformal Prediction intends to predict confidence set predictions rather than point predictions. This paper studies to improve the performance when there is domain shift for testing. First, this paper proposes ECP, which addaptively controls the perdiction set size by utilizing the uncertainty evaluation of the base model itself. Second, being motivated by the concept of TTA, this paper uses entropy minimization for optimizing the uncertainty. Without label annotation, this paper shows good performances.

**Strengths:**

The preliminaries are well formulated.

**Weaknesses:**

Please refer to the questions below

**Questions:**

- What is resp. in the first paragraph of introduction section? It is not desirable to use undefined abberviation term without explaining it.
- Why there are cases when entropy optimization shows worse performance than the original ECP?
- The problem setting suggested by the authors cannot be solved by the foundation model, e.g. CLIP or LLMs?

---

> ### Author Response · Authors · 2024-11-19
>
> We are glad you acknowledge our method's good performance.
>
> 1. What is resp. in the first paragraph?
>
> Thank you for pointing this out. By 'resp.' here we mean 'respectively'. We will edit the draft to omit undefined abbreviations.
>
> 3. Why there are cases when entropy optimization shows worse performance than the original ECP?
>
> Test time adaptation by entropy minimization is effective in improving coverage and / or set sizes on all the datasets we study (almost always improves both). In a small number of edge cases, it may slightly increase set sizes relative to ECP. While entropy minimization has proven very effective, it still has some shortcomings which continue to be studied [1]. We are confident that improvements to the base TTA methodology will effectively translate to improvements on EaCP.
>
> 4. The problem setting suggested by the authors cannot be solved by the foundation model, e.g. CLIP or LLMs?
>
> Although powerful, models like CLIP and LLMs still significantly struggle with distribution shift. They cannot solve the problem we investigate. However, future works can certainly study the effect of our proposed method on these models.
>
>
> Please let us know if you have any outstanding questions or concerns.
>
> [1] Press, Ori, et al. "The entropy enigma: Success and failure of entropy minimization." arXiv preprint arXiv:2405.05012 (2024).

---

### Official Review · Reviewer_rzHG · 2024-11-05

**Soundness:** 2
**Presentation:** 3
**Contribution:** 1
**Rating:** 3
**Confidence:** 4

**Summary:**

This paper proposes a novel conformal prediction method under distribution shift without using any labels from a shifted distribution. The key idea is rescaling a scoring function based on entropy, which measures a degree of shifts. Also, this paper updates the scoring function using test-time adaptation to enhance the efficiency of conformal sets. The proposed methods are evaluated over variations of ImageNet for distribution shifts.

**Strengths:**

This paper attacks an important problem of learning conformal sets under distribution shifts. The observation is simple and intuitive. It demonstrated the efficacy of the proposed method by using at least 6 shifted datasets.

**Weaknesses:**

I think the main concern of this paper is that the proposed method does not provide any theoretical guarantee, which is the crux of conformal prediction. Including this, see the following for additional concerns.

* The proposed method is heuristic, which does not align with paper trends of conformal prediction. Can you provide a theoretical coverage guarantee (i.e., the coverage probability bound) of the proposed method under the covariate shift assumption?
* For achieving a desired coverage, the method introduced two tuning knobs: \beta, f. \beta set to 1-\alpha, but choosing them is anyway very tricky in distribution shifts without labels. This suggests the proposed method is less practical. Could you suggest an adaptation method to learn f by only using unlabeled examples?
* In demonstrating the efficacy of the proposed method, it would be great if oracle results and no-shift results are included. In particular, you can consider SplitCP with access to labeled examples from a shifted distribution – this is an oracle result, which forms a empirical desired coverage / set size for the proposed method. Also, under no shift, the proposed method should work – this is also demonstrated to show the efficacy of the method as we don’t know whether there is a shift or not.
* An important baseline is missing. The main competitor of this paper is conformal prediction under covariate shift (i.e., Tibshirani et al., 2019). The comparison should be conducted. Could you apply your method to visualize the second plot with oracle weights in Figure 2 of Tibshirani et al. (2019) and compare an empirical coverage along with average prediction set size?

**Questions:**

* Can you provide a coverage guarantee of the proposed method?
* Can you add oracle results and no-shift results to justify the efficacy of the proposed method?
* Can you compare the proposed method with Tibshirani et al., 2019?

---

> ### Author Response · Authors · 2024-11-19
>
> Thank you for your insightful review! We are glad you appreciate the importance of this problem.
>
> 1. Theoretical coverage guarantee based on covariate shift assumptions?
>
> We agree that the coverage rate guarantee has been a popular trend within the field of conformal prediction, therefore, our paper is a bit different due to the absence of such guarantees. However, here we would like to challenge this convention by providing some alternative perspectives:
>
> - We would like to kindly re-emphasize that **the challenging setting we study (distribution shifts without test time labels) inherently suffers from a lack of non-trivial guarantees,** yet is quite common in practice.
>
>     To our knowledge, all existing attempts on this problem (discussed in Sec 1 and 2) make stringent assumptions or simplifications in order to prove things. For example, Tibshirani et al. (2019) studied CP under only covariate shift, however, accurately estimating the required likelihood ratio for that approach to work can be challenging in practice. Alternatively, the idea of Cauchois et al (2024) is to take a robust optimization perspective by assuming a certain maximum level of distribution shift and protecting against the worst case. However, we show that their guarantees fail in practice and achieve lower coverage than our ECP/EaCP on all of the natural distribution shifted datasets. This result highlights that the assumptions they require for their guarantee are difficult to satisfy in practice. Thus, without any assumptions of this type on the nature of the covariate shift, we believe our problem is ill-posed in the strict theoretical sense, such that any nontrivial coverage guarantee is impossible.
>
> In light of such limitations, we take a new perspective on this problem.
>
> - Instead of attempting to provide assumption-laden guarantees that won't hold in the real world, we instead aim for **practical approaches that consistently perform well**. Thus, while we do leverage CP-inspired techniques, we ultimately focus on providing ***general prediction sets*** that are accurate and efficient.
>
>
> Overall, we very much agree with you that coverage guarantees are an interesting aspect of conformal prediction. However, the reality is that machine learning systems must often make predictions or decisions, perhaps unwittingly, even in this extremely challenging setting of unlabeled distribution-shifted data. Ultimately, conformal prediction is simply one of many possible set-valued classification methods. We hope reviewers would acknowledge and appreciate our effective and practical approach as an important stepping stone towards addressing the issue of accurate and effective prediction sets in a generalizable and scalable manner.
>
> (1/2)

---

> ### Author Response · Authors · 2024-11-19
>
> 2. On hyperparameters and learning *f*.
>
> We would like to emphasize that we did essentially *zero hyperparameter tuning* in our paper. Although we do introduce two hyperparameters, we demonstrate that simply setting $\beta=1-\alpha$ works well in practice across many datasets, severity levels, error rates, as well as both stationary and continuous shifts.
>
> We do agree with you that learning $f$ is a fruitful area for further exploration, which we and hopefully others will continue to develop in future studies. In our current submission, we emphasized the development and thorough empirical validation of our novel ECP and EaCP methods. We demonstrate that a simple quadratic / linear scaling function works remarkably well on a large number of diverse datasets in this challenging setting. There are certainly a number of ways the choice of scaling function can be improved, and several recent works may potentially offer insights to this end [1,2,3], which we are eager to explore in future works.
>
> That said, the simplicity and effectiveness of 'off-the-shelf' quadratic or linear scaling functions is a key strength of our approach. While dataset-specific optimization might marginally improve performance, it would compromise the elegant simplicity of our method.
>
>
> 3. Oracle and no-shift results.
>
> Thank you for the suggestion! We are currently running these experiments and will provide the results shortly. We would also like to point out that due to our design choice in Equation 4, prediction sets will never be decreased and thus coverage cannot decrease including on in-distribution data.
>
>
> 4. Tibshirani et al. (2019) baseline comparison.
>
> While Tibshirani et al. (2019) is certainly a seminal work in this area, we respectfully disagree that this is a relevant baseline. This is primarily due to the fact that it is highly challenging to estimate accurate and reliable likelihood ratios for high-dimensional image classification tasks. In fact, we are not aware of any previous works that use the method from Tibshirani et al. (2019) in large scale image classification tasks. This sentiment is echoed in Cauchois et al. (2024), who similarly present several reasons why Tibshirani et al.'s approach is quite restrictive and not applicable to our setting.
>
> Instead, we focus on comparing against works that have shown positive results on ImageNet-scale tasks. This includes Cauchois et al., as well as **five other baselines** that have previously demonstrated strong performance on ImageNet. Note that these previous works also do not compare against Tibshirani et al. We hope it is clear now why we do not include this comparison.
>
> We hope this addresses your concerns, and we are glad to answer any further questions.
>
> [1] Miller, John P., et al. "Accuracy on the line: on the strong correlation between out-of-distribution and in-distribution generalization." International conference on machine learning. PMLR, 2021.
>
> [2]  Kim, Eungyeup, et al. "Reliable Test-Time Adaptation via Agreement-on-the-Line." arXiv preprint arXiv:2310.04941 (2023).
>
> [3] Kang, Katie, et al. "Deep neural networks tend to extrapolate predictably." ICLR 2024.
>
> (2/2)

---

> ### Comment · Reviewer_rzHG · 2024-11-26
>
> Dear Authors,
>
> Thanks for sharing your opinions.
> * I would appreciate novel viewpoints on building prediction sets under shifts, but I believe we have to find minimal assumptions that provide a guarantee under the distribution shift. Authors claimed that it is not easy to find a paper that provides guarantees with practical assumptions. But, I found the following paper (https://arxiv.org/abs/2106.09848) that carefully estimates the likelihood ratio, different to Tibshirani et al. (2019), suggesting that finding practical assumptions for a guarantee is possible.
> * Comparison with Tibshirani et al. (2019) should be conducted. You can just use an estimated likelihood ratio.
> * I want to see “Oracle and no-shift results”, but they are still missing.
>
> Because of these reasons, I’ll maintain my score.

---

> > ### Author Response · Authors · 2024-11-27
> >
> > Dear reviewer,
> >
> > Thank you for your response and pointing us to Park et al. ICLR 2022. It is a very relevant paper to our work as it is also concerned with unsupervised domain adaptation and prediction sets. While they present an interesting approach, we note that their setting of domain adaptation differs from ours as we consider both stationary and continuously shifting distributions. We would need to spend more time with this paper to understand its implementation and we appreciate that the authors have provided PyTorch code on GitHub.
> >
> > We have conducted the experiments you suggested using oracle sets and in-distribution data, and have included the results in Appendix D.5 and D.6 of the latest manuscript (PDF). The results of our experiments demonstrate that our ECP and EACP methods achieve set sizes similar to, and often better than, the oracle. In fact, our methods often produce sets substantially smaller than the oracle due to the entropy minimization. We further show that in-distribution coverage and set sizes are slightly conservative but nonetheless comparable to standard split conformal.
> >
> > In Appendix D.7, we present a comparison with the method developed by Tibshirani et al. (2019) using the ImageNet-V2, ImageNet-R and ImageNet-A distribution-shifted test sets. We tried to include the best implementation we could, however still emphasize the challenge this method faces in our setting, exacerbated by the sparsity of previous implementations at this scale. Thus, while further studies are required here, we are happy to include this result for completeness and see future work continue to explore this approach. Finally, we note the important point that their method is **ill-suited for continuous shifts.** In contrast, **we demonstrate the efficacy of our method on random and continuously shifting distributions**, further highlighting the benefit of our approach compared to existing works. Thus, we again emphasize our comparison with a large number of strong baselines that *have shown positive results* in our setting.
> >
> > We hope we were able to adequately address your questions and concerns in our rebuttal. We'd be happy to address any further questions you might have. If there are no further outstanding concerns, we kindly ask if you can please consider raising your score to reflect this. Thank you!

---

### Official Review · Reviewer_2xjb · 2024-11-05

**Soundness:** 2
**Presentation:** 3
**Contribution:** 2
**Rating:** 3
**Confidence:** 4

**Summary:**

The paper presents a novel approach to enhance the robustness of set-valued predictions, specifically Conformal Prediction (CP), in the face of distribution shifts without relying on test-time labels. The authors introduce two primary methods, ECP (Entropy scaled Conformal Prediction) and EACP (Entropy base-adapted Conformal Prediction), which leverage the uncertainty revealed by the base model to adjust prediction sets accordingly. Through extensive experiments on large-scale datasets and various neural network architectures, the paper demonstrates that these methods can significantly improve upon existing baselines and nearly match the performance of fully supervised methods.

**Strengths:**

The investigation presented in this paper on Conformal Prediction (CP) under distribution shifts is of considerable value, addressing an area that has received relatively little attention in existing studies.

**Weaknesses:**

The motivation behind this paper is that, under distribution shifts the overall output confidence of models tends to decrease, thereby affecting the performance of CP. However, there are several aspects in the method design that currently fail to convince me, such as:

1. Figure 1 observes the relationship between the model's output confidence and entropy, using this relationship as a key observation to guide the method design. But isn't this correlation trivial? Entropy and confidence are both indicators of model output uncertainty, especially when considering average entropy and confidence, as the method proposed in [1] treats both as indicators of model confidence.

2. The authors, through preliminary experimental observations, designed a method to enlarge the model's output under Out-Of-Distribution (OOD) conditions. I believe caution is needed for this design, as **existing research has widely pointed out that models face a more severe over-confidence phenomenon under OOD settings compared to In-Distribution (ID) settings**, and continuing to enlarge model outputs will inevitably exacerbate this issue. From this perspective, is the method in this paper reliable? Would a more over-confident model (which is certainly also more miscalibrated) be beneficial for CP?

3. Why do the authors use the quantile mechanism shown in Equation (3) to enlarge the model's output? This seems to lack insight. Assuming that the strategy to enlarge model confidence is correct (however, as mentioned in point 2, this assumption may not be correct), shouldn't we directly use a method like temperature scaling to align the model's average output on OOD and ID datasets?

I believe there are still many aspects of this paper that fail to fully convince the readers and require further clarification from the authors.

[1] Leveraging unlabeled data to predict out-of-distribution performance, ICLR 2022.

**Questions:**

Please refer to weaknesses.

---

> ### Author Response · Authors · 2024-11-19
>
> Thank you for your review, we are glad you acknowledge the importance of our problem setting as well as the extensive experiments and "significant improvement" of our methods. Regarding your comments:
>
> 1. On the relationship between the model's output confidence and entropy in Figure 1
>
> With all due respect, we believe that the reviewer has misinterpreted what is plotted in Figure 1. As indicated by the y-label, it plots *softmax value of the true label*, and it **does not** plot model confidence on the y-axis. The relationship between softmax of the true label and entropy is non-trivial. If this was instead "max softmax" (a common heuristic for confidence and the metric used in your cited paper), then we agree it would be trivial.
>
> 2. Model confidence under OOD settings
>
> We are aware of the issue of poorly calibrated models and agree it is an important concern (and related to model confidence). However, the connection the reviewer draws here may be related to the misunderstanding mentioned above.  Model calibration is tangential to our primary topic of constructing accurate and useful prediction sets. ECP only enlarges the *prediction set*, quite the opposite of producing a "more over-confident model" as a larger set conveys greater uncertainty. No model parameters change. The top-k predictions will not change, so standard calibration checks (e.g. ECE) would not be affected. For EaCP we perform TTA so model parameters do change, however, there is no evidence that the TTA procedure disrupts model calibration.
>
> >  Over-confidence on OOD
>
> We would like to point out that seminal papers on test-time adaptation (TTA) have established the reliable correlation between prediction entropy and distribution shift severity [1]. Our results also corroborate this phenomenon. In this context, it is unclear what is meant by "continuing to enlarge model outputs will inevitably exacerbate this issue." Please refer to our earlier point about larger confidence sets conveying greater *uncertainty* rather then overconfidence.
>
> 3. Why use the quantile mechanism.
>
> As mentioned in our paper on line 248, we use the fact that quantiles are generally more robust than expectations. Certainly other choices, such as temperature scaling, may exist and we would be excited for future works to build on this. Nonetheless, we hope to have demonstrated that our quantile-based approach works well on diverse datasets and distribution-shift settings as an initial approach.
>
> We hope this addresses your concerns, and we are happy to address any outstanding questions!
>
>
> [1] Tent: Fully Test-time Adaptation by Entropy Minimization. ICLR 2021.

---

### Author Response · Authors · 2024-12-03
**General Response**

Dear reviewers,

Thank you for your comments and reviews of our work.

We are glad that Reviewer 2xjb acknowledges the significant improvement upon baselines of our method, and that the setting we investigate is both important and currently under-studied in the literature. The importance of our setting and efficacy of our method are also echoed by Reviewer rzHG and Reviewer cR3g.

We hope that our response has clarified the Reviewer 2xjb's concerns. Importantly, the core concern of over-confidence under OOD is addressed by the fact that our method **enlarges prediction sets under OOD data thus conveying *greater* uncertainty**. We also point to Figure 4 showing that the varying set sizes of our methods starkly contrast with the static sets of other baselines; this is indeed the desired behavior the reviewer mentioned.

We have further updated our manuscript to contain the additional experiments suggested by Reviewer rzHG, demonstrating our method outperforms Tibshirani et al. (2019), and that it is comparable (and often better than) the Oracle in terms of prediction set sizes. We also appreciate their openness to "novel viewpoints on building prediction sets under shifts." While statistical guarantees may be sought after in some settings, we strongly emphasize the **immediate practical improvements** of our methods and significant improvements upon existing baselines that practitioners and researchers can already begin using. We certainly hope that future work can further expand the theoretical motivations of our approach, as has often been the case in ML research.

Finally, we believe our response has adequately addressed all of Reviewer cR3g's stated weaknesses, including their questions regarding foundation models and edge cases where entropy adaptation has room for improvement.

Thank you!

---

### Meta-Review · Area_Chair_Bhgo · 2024-12-17

**Metareview:**

This paper investigates an interesting problem where the goal is to improve the practical performance of conformal prediction using only unlabeled data from the shifted test domain. To solve this problem, this paper proposes two new methods, whose main idea is to adjust the score function in conformal prediction according to its base model's own uncertainty evaluation. Experiments show that the proposed methods outperform existing baselines.

Pros:
- The investigated problem is interesting.
- The effectiveness of the proposed methods is demonstrated.

Cons:
- The motivation of the method design is not so convincing.
- No theoretical guarantee is provided for this conformal prediction study.

The above two defects of this paper are quite signficant and cannot be ignored for the research on conformal prediction. Therefore, I would like to recommend rejecting this paper.

**Additional Comments On Reviewer Discussion:**

This paper receives the scores of 3 (Reviewer 2xjb), 3 (Reviewer rzHG), 5 (Reviewer cR3g).

Reviewer 2xjb raised the main concern that the motivation of the method design is not so convincing, and pointed out several aspects. The authors' rebuttal partially addressed this concern.

Reviewer rzHG raised the main concern that no theoretical guarantee is provided for this conformal prediction study. Reviewer rzHG also rased other concerns, while the authors' rebuttal has addressed them and included the modifications in the appendix.

Reviewer cR3g has several minor concerns, but I feel that the authors could provide more explanations or justifications. For example, CLIP can also be used to do classification, but why CLIP cannot be the base model in this paper?

Considering the above situations, I recommend rejection.

---

### Decision · Program_Chairs · 2025-01-22

Reject